# Mechanisms of Resistance to Antibody–Drug Conjugates

**DOI:** 10.3390/cancers15041278

**Published:** 2023-02-17

**Authors:** Rachel Occhiogrosso Abelman, Bogang Wu, Laura M. Spring, Leif W. Ellisen, Aditya Bardia

**Affiliations:** Massachusetts General Hospital, Harvard Medical School, Boston, MA 02114, USA

**Keywords:** antibody–drug conjugates, breast cancer, targeted therapies

## Abstract

**Simple Summary:**

Antibody–drug conjugates (ADCs) are a growing class of therapies that aim to delivery therapy more efficiently, with fewer side effects, than conventional chemotherapy. ADCs are composed of an antibody linked to a chemotherapy payload, allowing targeted delivery of the chemotherapy. In the last decade, several antibody–drug conjugates have improved treatment options in breast cancer. However, patients usually progress on these agents, and more research is needed into why this resistance occurs. Given the complex structure of antibody-drug conjugates, resistance may be related to changes in antigen expression, ADC processing, and the chemotherapy payload. This paper reviews the literature on the mechanisms of resistance to antibody–drug conjugates including pre-clinical and clinical studies in breast cancer and other malignancies. This review includes information on ADCs that have been approved for use in breast cancer and ADCs in development that seek to overcome the proposed mechanisms of resistance to improve treatment options for patients.

**Abstract:**

Antibody–drug conjugates (ADCs), with antibodies targeted against specific antigens linked to cytotoxic payloads, offer the opportunity for a more specific delivery of chemotherapy and other bioactive payloads to minimize side effects. First approved in the setting of HER2+ breast cancer, more recent ADCs have been developed for triple-negative breast cancer (TNBC) and, most recently, hormone receptor-positive (HR+) breast cancer. While antibody–drug conjugates have compared favorably against traditional chemotherapy in some settings, patients eventually progress on these therapies and require a change in treatment. Mechanisms to explain the resistance to ADCs are highly sought after, in hopes of developing next-line treatment options and expanding the therapeutic windows of existing therapies. These resistance mechanisms are categorized as follows: change in antigen expression, change in ADC processing and resistance, and efflux of the ADC payload. This paper reviews the recently published literature on these mechanisms as well as potential options to overcome these barriers.

## 1. Introduction

Antibody–drug conjugates (ADCs), novel agents that use selective targeting by monoclonal antibodies to deliver cytotoxic chemotherapies and other bioactive payloads to cells expressing a particular antigen, seek to deliver highly potent agents while minimizing off-target toxicity. In recent years, successful trials of ADCs in breast cancer and other malignancies have demonstrated that ADCs can be effective and limit toxicity, in some cases supplanting traditional chemotherapy. While these agents have had marked success, particularly in the metastatic setting, almost all advance-stage patients treated with ADCs develop resistance. Efforts to characterize and define the mechanisms to explain resistance have incorporated both pre-clinical models and clinical investigation, spanning genomics to proteomics and even direct visualization of resistant cell lines. Altogether, recent discoveries of the processes mediating resistance provide hope for new innovations that expand the therapeutic window of these highly active therapies.

## 2. Mechanism of Action

ADCs are composed of an antibody targeting an antigen associated with malignancy, combined through a linker with a cytotoxic moiety, allowing for a more specific delivery of chemotherapy. The ADC binds a receptor on the target cell and is then internalized into the cell through receptor-mediated endocytosis. Once inside the cell, the antibody and cytotoxic agent are separated in the lysosome, where the linker is cleaved, releasing the payload. Targeting cells via antibodies aims at specifically delivering the payload while minimizing toxicity, enabling the use of chemotherapy as much as 100 to 1000 times more concentrated than standard cytotoxic chemotherapy [1].

The antibody selected for use in ADCs is often immunoglobulin G (IgG), most often the IgG1 subclass. Antigen targets are ideally ubiquitous in malignant cells but are rarely found in normal cells, to limit off-target toxicity [2]. Antibodies are connected to payloads through novel linkers that vary in properties, in ways that can be manipulated for therapeutic use. Linkers can be non-cleavable, requiring additional processing to release their payload, or cleavable, in the event of tumor-specific factors such as a change in pH or enzyme. Cleavable linkers provide the advantage of a more efficient delivery of the chemotherapy payload and may be more likely to impact neighboring antigen-negative cells, which may or may not be desirable. In contrast, non-cleavable linkers may deliver their payload with more specificity but may also need additional processing such as lysosomal degradation, which can impact the payload [3]. The chemotherapy agents used are typically more potent than can be tolerated in a conventional delivery. Agents of choice include auristatins, calicheamicins, maytansinoids, and camptothecin analogues [2].

ADCs can also be categorized by their drug-to-antibody ratio (DAR), a measure of the number of chemotherapy moieties conjugated to each antibody. In theory, ADCs with a higher DAR would be expected to be more potent, although in some situations drugs with higher DARs were found to have an increased hepatic clearance that may limit their efficacy [4,5]. ADCs were first demonstrated to have efficacy in settings where cancer was relapsed or refractory to standard chemotherapy treatments. The success in ADCs vs. standard therapy in these cases may be related to the increased heterogeneity in pretreated tumors. This efficacy may be mediated by the “bystander effect”, where tissues adjacent to those expressing the target antigen are also targeted by the cytotoxic payload [6,7]. This effect was observed in early studies of the ADC trastuzumab deruxtecan (T-DXd), where HER2- cells’ neighboring cells that were expressing HER2 were also targeted by the agent [1,6]. While this mechanism provides impact in heterogeneous tumors, given decreased specificity there is an increased risk of toxicity [2].

## 3. ADCs Approved in Breast Cancer

Trastuzumab emtansine, known as T-DM1, was the first antibody–drug conjugate approved for use in breast cancer. In the TH3RESA trial, T-DM1 demonstrated an improved PFS (6.2 months vs. 3.3 months, HR 0.528) and later overall survival compared to the treatment of physician’s choice for patients with HER2+ breast cancer who had already previously received two prior lines of chemotherapy including a taxane in any setting and trastuzumab/lapatinib for an advanced disease [8]. Notably, T-DM1 retained activity against tumors that had progressed on prior HER2-directed therapy, an early indication that antibody–drug conjugates may have renewed efficacy against previously exploited targets. In the EMILIA study, HER2+ metastatic breast cancer previously treated with trastuzumab and a taxane demonstrated an improved PFS (9.6 vs. 6.4 months, HR 0.68) compared to lapatinib/capecitabine and also demonstrated an improved overall survival (30.9 months vs. 25.1 months, HR 0.68, *p* < 0.001). This result elevated T-DM1 to a second-line therapy for metastatic HER2+ disease [9]. While early studies demonstrated promise, there were some limitations to the success of the agent, as seen in the MARIANNE study, where T-DM1 with and without pertuzumab was not superior to the combination of trastuzumab and a taxane as the first-line therapy for metastatic breast cancer [10]. In the KRISTINE trial, T-DM1 and pertuzumab (P) were less effective in leading to a pathologic complete response (pCR) than TCHP for neoadjuvant use in early-stage/local HER2+ breast cancer [11]. Notably, 44% in the T-DM1+P arm still achieved pCR without the use of cytotoxic chemotherapy, suggesting some role for this type of de-escalation in the future. Patients who had a progression prior to surgery were more likely to have heterogeneous HER2 expression, suggesting that patients with a high/more homogenous HER2 staining could be selected for targeted de-escalation in a different setting [12]. Finally, the KATHERINE trial demonstrated the effective use of T-DM1 for patients with residual HER2+ disease after neoadjuvant use of trastuzumab and a taxane, with invasive disease found in 12.2% of the cohort that had received 14 cycles of T-DM1 after surgery, compared to 22.2% of the group that received trastuzumab after surgery [13].

The next antibody–drug conjugate approved for use in breast cancer was sacituzumab govitecan (SG), composed of a monoclonal antibody targeting TROP-2 that is connected via a cleavable linker to SN-38, an irinotecan metabolite [14] (Figure 1). SG obtained approval based on the ASCENT trial, a phase III trial comparing SG vs. single agent chemotherapy in patients with metastatic triple-negative breast cancer (TNBC) who had received two prior lines of chemotherapy. Patients treated with SG were found to have an improved PFS of 5.6 months compared to 1.7 months for those receiving the treatment of physician’s choice, along with an improved median overall survival of 12.1 months vs. 6.7 months in the TPC (treatment of physician’s choice) group [15]. As a result of these data, SG received full FDA approval for treatment of mTNBC as a second-line treatment, the first approved ADC for metastatic mTNBC [16], and in 2023 also received approval for patients with metastatic Hormone Receptor Breast cancer.

Finally, trastuzumab deruxtecan (T-DXd) was most recently the second antibody–drug conjugate approved for use in HER2+ advanced breast cancer and has also defined a new class of “HER2-low” disease. T-DXd is composed of trastuzumab combined via a cleavable linker with deruxtecan, a topoisomerase-I inhibitor. This compound has a relatively high drug:antibody ratio of 8:1, a property along with the linker structure that may explain its efficacy in patients with heterogeneous HER2 expression [17,18,19]. T-DXd was studied in a series of trials known as the DESTINY-Breast trials, with many still ongoing. In DESTINY-Breast 01, T-DXd demonstrated a confirmed response of 60.9% in patients with metastatic HER2+ breast cancer previously treated with T-DM1, with a median progression-free survival of 16.4 months [20]. T-DXd was compared directly to T-DM1 in DESTINY-Breast 03, demonstrating an overall response rate of 79.7% in the T-DXd group vs. 34.2% in the T-DM1 group, with a median PFS that was unable to be calculated due to ongoing treatment in the T-DXd group, compared to 6.8 months in the T-DM1 group [21]. As noted above, T-DXd demonstrated efficacy in clinically HER2-negative patients with heterogeneous HER2 expression, effectively creating a new clinical categorization of patients with HER2-low disease (IHC 1+ or IHC 2+ without FISH amplification). In the DESTINY-Breast 04 trial, patients classified as HER2-low who, like above, had metastatic disease and had previously received 1–2 lines of therapy demonstrated an improved response to T-DXd compared to TPC, with a median PFS of 9.9 months vs. 5.1 months in TPC, and OS of 23.4 months in the T-DXd group vs. 16.8 months in the TPC cohort [22].

## 4. Mechanisms of Resistance 

Potential mechanisms of resistance can be categorized based on the complex structure of antibody-drug conjugates. Reviewed below, and summarized in Table 1, are proposed preclinical and clinical resistance mechanisms categorized by changes in antigen expression, processing of the antibody drug-conjugate, and changes in the chemotherapy payload. Altogether, categorizing resistance mechanisms with this approach provides direction for future research to further understand these mechanisms and new targets for drug development that can expand the efficacy of antibody-drug conjugates. 

### 4.1. Antigen Expression

In early trials of T-DM1, it was observed that tumors with a higher and more homogenous HER2 expression were more likely to respond to therapy. Given that HER2 can have quite heterogeneous expression, with intratumor heterogeneity reported in 16–36% of cases, agents that require a uniformly higher HER2 expression would meet resistance in the face of any change in the HER2 levels. Further evidence for this hypothesis includes observations that HER2+ tumors demonstrate a lower HER2 expression after treatment, and more heterogeneous expression is associated with a higher relapse rate and lower rates of survival [23]. A study of early stage HER2+ breast cancer patients treated with neoadjuvant T-DM1 and pertuzumab found that the pre-treatment presence of HER-2 heterogeneity, defined as ERBB-2 amplification in 5–50% of tumor cells or an area that was HER2-FISH negative, was inversely predictive of treatment response. In fact, among those with heterogeneous pre-treatment biopsies, 0% achieved pathologic complete response, while 55% of non-heterogeneous patients were found to have pCR with the combination of T-DM1 and P [24,25]. Additionally, the circulating tumor DNA (ctDNA) of patients with T-DM1 resistance was shown to have tumor cells with less HER2 amplification [12,26]. Further evidence for the hypothesis that decreased antigen expression can mediate ADC resistance came from preclinical cell lines. JIMT1 cells, resistant to trastuzumab, were used to create xenograft tumors that responded to T-DM1 at high concentrations. These tumors were then treated with cyclical T-DM1 to create resistance, and subsequent testing demonstrated decreased HER2 expression [27,28]. A preclinical example suggesting the importance of the bystander effect in overcoming decreased antigen expression comes in a model of gastric cancer cells. These cells were made resistant to T-DM1 and then demonstrated resistance to ADCs with non-cleavable linkers, while retaining sensitivity to ADCs with cleavable linkers and cell-permeable payloads [29]. A similar study with the JIMT1 cell line established that restoring HER2 expression reversed resistance to T-DM1 [28].

In addition to aberrant antigen expression level, dimerization of an antigen with another cell surface receptor is potentially able to mediate resistance to ADCs. NRG-1β, a ligand known to elicit HER2/HER3 heterodimerization, suppresses the cytotoxic activity of TDM-1 in a subset of HER2-amplified breast cancer cell lines. Such resistance can be overcome by adding pertuzumab, a monoclonal HER2 antibody that blocks HER2/HER3 dimerization and downstream signaling. The combination of TDM-1 and Pertuzumab showed a synergistic effect in both in vitro and in vivo tumor xenograft studies [30]. Another strategy utilizes the biparatopic antibodies in ADCs, which involves two arms that recognize different epitopes of the target antigen. This demonstrated success in preclinical studies of REGN5093-M114, a biparatopic MET-targeted ADC that is under investigation in MET-overexpressed EGFR-mutant NSCLC cell lines [31].

Trastuzumab deruxtecan, with important structural differences from T-DM1 including a membrane-permeable payload and a cleavable linker, demonstrated efficacy against tumors that had become resistant to T-DM1. It is hypothesized that this resistance to T-DM1 was overcome by mechanisms including the bystander effect, which was mediated by the aforementioned properties of T-DXd, allowing the drug to penetrate cells adjacent to tumor-expressing target cells that have evolved resistance through decreased HER2 expression [12,23]. This potential mechanism may explain some of the early findings of the DAISY trial, where T-DXd was investigated in patients with metastatic breast cancer with varying HER-2 expression, with cohort 1 including patients with traditional HER2+ disease (IHC 3+ or IHC 2+/FISH+), cohort 2 including patients with HER2 low disease (IHC 2+, FISH negative, or IHC 1+), and cohort 3 including patients with IHC 0 (HER2-null). Evidence of clinical activity was seen after T-DXd was administered in all three cohorts, including the HER2-null group, although the duration of the response and PFS appeared to be longer in patients with higher HER2 expression [32].

Given the impact of antigen heterogeneity in mediating resistance to ADCs, future strategies to address this resistance could include agents with dual antibodies (bispecific ADCs). In addition, combination therapies that increase expression on the antigen could be valuable. In a study of ERBB2-amplified or -mutant lung cancers, co-treatment of lung cancer cells with T-DM1 and the irreversible pan-HER kinase inhibitors neratinib increased uptake of T-DM1, but when the reversible HER2 inhibitor lapatinib was used, a decreased uptake of T-DM1 was seen [33].

### 4.2. ADC Processing

The complexity of ADCs, particularly compared to small molecules, provides many possible opportunities for resistance to emerge (Figure 2). In many cases, particularly in ADCs with non-cleavable linkers, before an ADC releases its payload, it typically binds the target antigen, is internalized into the cell, and is processed intracellularly. Some causes of resistance and, particularly, the intracellular mechanisms are difficult to demonstrate as a driver of resistance in pre-clinical studies, but inferences have been drawn based on observations of therapy-resistant models [27]. The hypotheses that have been proposed for mediating resistance to ADCs involve alterations in intracellular uptake and processing. Before the ADC even enters the cell, one proposed mechanism involves the decreased penetration into the cell by barriers, including an increase in the cellular basement membrane [27]. Another proposed mechanism comes from recent preclinical work demonstrating that abnormal endosomal transit may be involved in T-DM1 resistance [25]. One mechanism by which ADCs ensure specificity is via clathrin-mediated endocytic uptake in the cell expressing the target antigen. In the N89-TM cell line that was made resistant to T-DM1, an alternative mechanism for uptake was discovered, where cells used caveolin-1 (CAV1)-coated vesicles, which may be less efficient [17,34]. In keeping with this observation, a recent study in gastric cancer found a negative correlation between tumor CAV1 level and T-DM1 tumor uptake. Genetic or pharmacologic inhibition of CAV1 increased T-DM1 uptake and synergized with T-DM1 in vivo using multiple xenograft models [35]. In addition, glycosaminoglycan modification negatively regulates the internalization of tumor antigen CAIX and anti-CAIX ADCs by promoting the association between CAIX and the CAV1 in membrane lipid rafts. Pharmacologically inhibiting protein glycosaminoglycan modification increased anti-CAIX ADC internalization and cytotoxic activity [36]. In a different study, cells resistant to T-DM1 were noted to have an absence of proteolytic activity once the ADC reached the lysosome due to a change in pH, leading to an accumulation of the agent mimicking lysosomal storage diseases [17,34,37].

The JIMT1-TM cell line, resistant to T-DM1, was found to have an increase in proteins including Rab5B, ATG9a, and HTT, which mediate lysosomal processing and the transportation of vesicles. Proteomic analysis revealed particularly high levels of proteins including Rab 6, a protein involved in microtubule-mediated transport, and the cytoskeleton-involved protein PAK4 [28,38,39]. To expand upon these findings, JIMT1-TM cells along with the 361-TM cell line were observed under live cell imaging microscopy. For the 361-TM cell line, compared to the parent cells that were not resistant to T-DM1, ADCs with non-cleavable linkers spent a significantly longer time in the lysosome, raising concern about less efficient ADC processing. Interestingly, this finding was not observed in ADCs with cleavable linkers, suggesting a mechanism for overcoming resistance. In the JIMT1-TM cell line, both cleavable and non-cleavable linkers demonstrated longer co-localization in the lysosome than in the parent cells. Unsurprisingly, both resistant cell lines were found to have decreased linker–payload metabolites compared to the parent cells, although it is uncertain whether this is driven by decreased HER2 expression, decreased ADC processing, or both [28]. In line with the notion that payload release may involve different ADC trafficking requirements depending on the linker status, a recent study using CRISPR screening demonstrated that cleavable linker ADCs are rapidly processed immediately after ADC uptake and trafficked into the early endosome, while non-cleavable linker ADCs require further lysosomal delivery to successfully release the payload. Additional targeted library screening identified that the depletion of sialic acids sensitizes cells to ADC toxicity by boosting ADC lysosomal delivery, potentially by reducing ADC recycling [40].

### 4.3. Payload

Some observed mechanisms of resistance involve the payload itself and may be overcome by use of an ADC with an alternate payload. For example, resistance to ADCs with a payload of topoisomerase inhibitors can be driven by changes in the expression of topoisomerase or changes in the downstream signaling mechanisms [27] (Figure 3). This has been observed in tumor models of non-Hodgkin’s lymphoma, where changing ADCs incorporating an auristatin-based payload to those with an anthracycline-based payload led to a further response to ADC-based therapy [17,41]. Similarly, in breast cancer cells that were resistant to T-DM1, the cells remained susceptible to DS-8201a (T-DXd), and these cells experienced less than half of the viability compared to cells treated with a control compound. To follow up on these results, tumor xenografts in mice were created from cells that were both sensitive and resistant to T-DM1. The xenografts created from T-DM1-resistant cells was unsurprisingly not sensitive to T-DM1, but when treated with DS-8201a, these mice demonstrated a reduced tumor volume [42]. Applications of this theory include the development of additional HER2-targeted ADCs, such as trastuzumab duocarmycin, containing a novel DNA-alkylating payload, and ARX788, which uses a non-cleavable linker and a payload designed for decreased cell permeability to minimize the toxicities associated with the bystander effect [25]. In addition, phase I/II studies are ongoing for SKB-264, another TROP-2-targeted ADC with a belotecan derivative (alternate topoisomerase-1 inhibitor), joining datopotamab deruxtecan and sacituzumab govitecan as targeting TROP-2 but presenting a payload alternative [43].

The ability to perform whole-exome sequencing on patients as they receive antibody–drug conjugate therapy allows for the ability to track the development of resistance mutations in real time and theorize strategies for future treatment. For example, previous work performed by our group evaluated the mechanisms of resistance in a patient with a prolonged response to SG (over 8 months) who had previously undergone whole-exome sequencing prior to initiating this therapy. After progression, rapid autopsy sampling was performed, and specimens before and after treatment were compared with SG. An analysis of the mutations that developed after treatment demonstrated a tumor subclone harboring mutations of *TOP1* (which encodes topoisomerase-1, the target of the SG payload), along with a distinct subclone harboring a mutation of *TACSTD2* (which encodes TROP2, the antigen target of SG). These findings imply there are parallel mechanisms of resistance that may develop against both the antibody and payload of ADCs. This finding has potential implications for the resistance to sequential ADCs including those with similar payloads (trastuzumab deruxtecan, a Topo-I inhibitor payload) or similar antibody targets (other TROP2 ADCs such as datopotamab deruxtecan) [44]. Future innovations in targeted and rapid sequencing offer the opportunity to provide personalized treatment directives to patients based on the resistance mutations they have developed from previous therapies.

**Table 1 cancers-15-01278-t001:** Categories of resistance mechanisms.

Resistance Mechanism	ADC Where This Has Been Seen	Cancer Type	Examples
Antigen expression	T-DM1	Breast cancer, lung cancer	Patients with more heterogenous HER-2 expression were less likely to achieve pCR with neoadjuvant T-DM1 and pertuzumab [24]. JIMT1 cells, resistant to trastuzumab, had decreased HER2 expression after being made resistant to T-DM1 [27,28].
ADC processing	T-DM1	Gastric cancer, breast cancer	Caveolin-1-coated vesicles may make cells uptake T-DM1 less efficiently. CAV-1 expression in gastric cancer has been associated with decreased T-DM1 uptake [17,34,35].
Payload	T-DM1, SG, ADCs with topoisomerase inhibitor payloads, gemtuzuman ozogamicin	Breast cancer, AML, non-Hodgkin’s lymphoma	AML cell lines resistant to the ADC gemtuzumab ozogamicin demonstrated a treatment response when treated with an ADC with a decreased ability to efflux [17]. Breast cancer cell lines that were resistant to T-DM1 retained sensitivity to T-DXd. Tumor-derived xenografts in mice that were resistant to T-DM1 demonstrated a decreased tumor volume when treated with T-DXd [42]. ATP-binding cassette transporters have increased expression (20–50× higher) in cells resistant to T-DM1 [42]. A patient who experienced a prolonged response to SG and then developed resistance underwent rapid autopsy biopsies after death. Sequencing of her pre- and post-treatment biopsies demonstrated that after SG, resistance mutations emerged effecting the response to the payload (topoisomerase-I inhibitor) as well as the antigen target (Trop-2) [44].

The increased efflux of the ADC payload, mediated by ATP-binding cassette transporters, is another proposed mechanism of resistance. The transporters were noted to have increased expression, up to 20–50× higher, in cells resistant to T-DM1 as opposed to the parent cells [42]. This mechanism was observed in models of AML cells as a means of resistance to gemtuzumab ozogamicin (GO). Similarly, patients with lower levels of ABCB1, which encodes an ATP binding cassette, had an improved response to GO. In a different cell line, there was increased expression of the gene ABCC1 after resistance to T-DM1 developed [27,28]. Finally, in preclinical models of paclitaxel-resistant cancer cell lines, resistance to ADCs with taxane-derived payloads was associated with the activation of *TUBB3* mediated by overexpression of ABCB1 [45,46].

Multiple possible preclinical and clinical mechanisms have been suggested to target these means of resistance. A cell line resistant to T-DM1 was found to demonstrate resistance against trastuzumab-directed, auristatin-based ADCs with non-cleavable linkers. When MRP1, which encodes the drug-efflux pump ABCC1, was reversed or knocked down with siRNA, the cells demonstrated a renewed sensitivity to non-cleavable ADCs [27,28]. Interestingly, in AML cells resistant to GO, when a cytotoxic agent that had a poor efflux potential, vadastuximab talirine, was used instead, there was a renewed response [17]. A preclinical breast cancer mouse model, treated with an anti-nectin-4-directed ADC known as N41mab- vcMMAE, was analyzed using RNAseq after 9 months of treatment with the ADC. Moreover, in this model, there was upregulation in ABCB1. When the ADC was administered in combination with the P-gp pharmacologic inhibitor tariquidar, rapid treatment response was seen, which was substantially better tolerated than the combination of tariquidar and docetaxel [47].

## 5. Conclusions

While the growth and widespread use of antibody–drug conjugates in breast cancer have provided patients with new treatment options, most tumors eventually develop resistance to these agents. Recent preclinical models and clinical analyses have sought to model these mechanisms of resistance with the goal of developing strategies to extend and improve the therapeutic use of these agents. Given the complexity of ADCs, resistance mechanisms have been described or suggested targeting different components of the ADC. Mechanisms involving decreased antigen expression, decreased ADC trafficking and processing, resistance to the cytotoxic payload, and the increased efflux of the agent out of the cell are all potentially targetable processes that could be used for future drug development. Future efforts to overcome the initial resistance to ADCs should build upon the described mechanisms. Ongoing research in optimizing sequential treatment of ADCs and combination therapies will hopefully shed additional light on best use of these agents. Altogether, these efforts exemplify the promise of translational research in prompting new treatment options and, potentially, extending the clinical outcomes observed with these therapies.

## Figures and Tables

**Figure 1 cancers-15-01278-f001:**
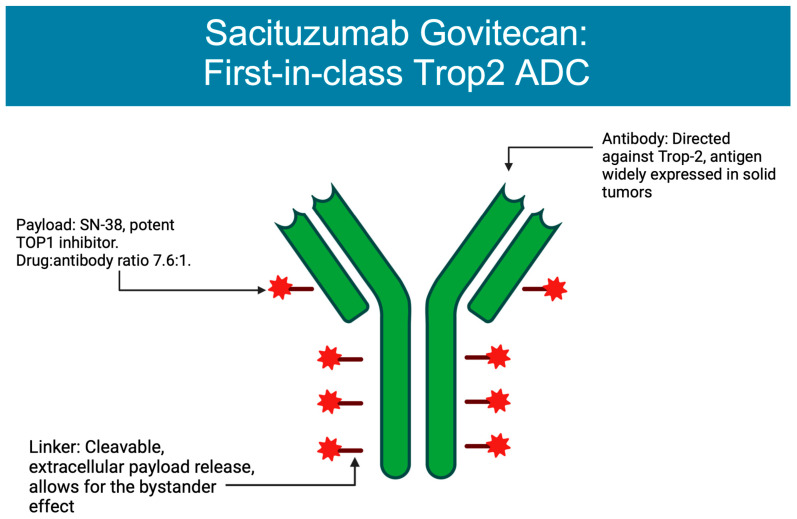
Structure of sacituzumab govitecan. Figure created on BioRender.

**Figure 2 cancers-15-01278-f002:**
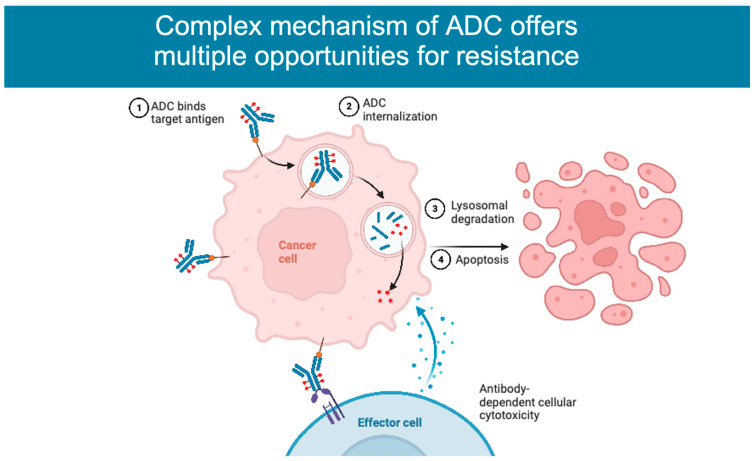
Possible resistance mechanisms based on ADC. Figure created on BioRender.

**Figure 3 cancers-15-01278-f003:**
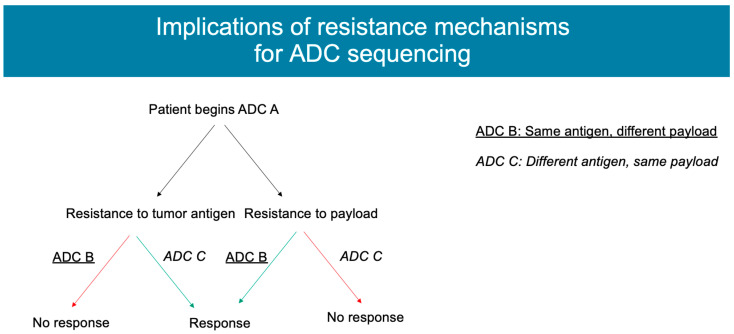
ADC resistance mechanisms. Figure created on BioRender.

## Data Availability

Not applicable.

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
