# Peer review of "Mechanisms of Resistance to Antibody–Drug Conjugates"

_cancers, 2023, doi:10.3390/cancers15041278_

Round 1

Reviewer 1 Report

This is a nice and conclusive review of the resistance mechanisms of ADCs in breast cancer. Minor comment to improve the article is to alter the Table 1 to present Resistant mechanism in 1st column, ADC in 2nd column, cancer type in 3rd column (as you mention gemtuzumab ozogamicin in AML) and a brief description with the ref in the 4th column. It will make it more clear to the reader and minimize all the wording. In addition, there is no mention of TDxd resistance mechanism/example. Finally, there is no mention of new ADCs (i.e trastuzumab duocarmazine, distamab vedotin, latiratumab vedotin) and their mode of action that would potentially help overcome resistance to the current ADCs.

Author Response

Thank you for your review! I have modified the table and added commentary on additional ADCs. 

Reviewer 2 Report

Antibody-drug conjugates (ADCs) are a rapidly expanding class of biotherapeutics that utilize antibodies to selectively deliver cytotoxic drugs to the tumor site. As of Jan 2022, the U.S. Food and Drug Administration (FDA) has approved ten, twelve ADCs, namely Adcetris®, Kadcyla®, Besponsa®, Mylotarg®, Polivy®, Padcev®, Enhertu®, Trodelvy®, Blenrep®, Zynlonta®, Tivdak® and Elahere™. The current approved ADCs include six directed against six different targets in hematological malignancies (CD33, CD30, CD22, CD79b, BCMA, CD19) and six targeting five different targets in solid tumors (HER2 (high and low), Nectin-4, TROP-2, tissue factor, Folate Receptor alpha).

As stated by the authors, resistance to ADCs is one of a limitation and have identified the main mechanisms (change in antigen expression, in ADC processing and resistance and efflux of the ADC payload). The recently published literature on these mechanisms is reviewed as well as the possible options to improve the design of next generation ADCs.

The manuscript is well written by experts in the field and deserves to be published in Cancers following brief discussions of the four 2023 related papers listed below:

·       Targeting HER2-positive breast cancer: advances and future directions. Sandra M. Swain, Mythili Shastry, Erika Hamilton. Nat Rev Drug Discov. 2023; 22(2): 101–126. Published online 2022 Nov 7. doi: 10.1038/s41573-022-00579-0

·       Preclinical Study of a Biparatopic METxMET Antibody-Drug Conjugate, REGN5093-M114, Overcomes MET-driven Acquired Resistance to EGFR TKIs in EGFR-mutant NSCLC. Oh SY, Lee YW, Lee EJ, Kim JH, Park Y, Heo SG, Yu MR, Hong MH, DaSilva J, Daly C, Cho BC, Lim SM, Yun MR. Clin Cancer Res. 2023 Jan 4;29(1):221-232. doi: 10.1158/1078-0432.CCR-22-2180.

·       Payload diversification: a key step in the development of antibody–drug conjugates Louise Conilh, Lenka Sadilkova, Warren Viricel, Charles Dumontet. J Hematol Oncol. 2023; 16: 3. Published online 2023 Jan 17. doi: 10.1186/s13045-022-01397-y

·       Antibody-drug conjugates in lung cancer: dawn of a new era? Coleman N, Yap TA, Heymach JV, Meric-Bernstam F, Le X. NPJ Precis Oncol. 2023 Jan 11;7(1):5. doi: 10.1038/s41698-022-00338-9.

Author Response

Thank you for your review! I have added all of the articles you suggested- please see the attachment.
